

# Upright walking has driven unique vascular specialization of the hominin ilium

Dexter Zirkle, Richard S. Meindl and C. Owen Lovejoy

Division of Biomedical Sciences and Department of Anthropology, Kent State University, Kent, Ohio, United States

## ABSTRACT

**Background:** A novel physis in hominins modulates broadening and shortening of the ilium. We report analysis of a vascular canal system whose origin may be associated with this physis and which appears to be also unique to hominins. Its presence is potentially identifiable in the fossil record by its association with a highly enlarged foramen that is consistently present in modern humans and hominin fossils.

**Methods:** We measured the diameter of this foramen in humans, fossil hominins, and African great apes and corrected for body size.

**Results:** The mean relative human foramen diameter is significantly greater than those of either *Pan* or *Gorilla*. Moreover, eight of the nine values of the *Cohen's d* for these differences in ratios are highly significant and support the ordering of magnitudes: *Pan* < *Gorilla* < *Homo*. The relative foramen diameter of A.L. 288-1 is above the 75th percentile of all other hominoids and at the high end of humans. The foramen is also present in ARA-VP-6/500.

**Conclusions:** We posit that the presence and significant enlargement of this foramen in fossils can reasonably serve as an indicator that its anterior inferior iliac spine emerged *via* the unique hominin physis. The foramen can therefore serve as an indicator of hominin iliac ontogenetic specialization for bipedality in fossil taxa.

## INTRODUCTION

The hominin ilium is strikingly superoinferiorly shorter than are the ilia of other primates, and its isthmus is mediolaterally much broader. We previously demonstrated that a unique physis, present only in hominins, likely guides these dimensional differences with other primates. That is, whereas only the acetabular rim exhibits an open growth plate during anterior pelvic growth in all other primates, in hominins a novel growth plate, the *Anterior Physis of the Ilium* (hereafter merely the "anterior physis") (*Zirkle & Lovejoy, 2019*) progressively spreads superiorly away from its acetabular portion to occupy much of the ilium's anterior edge between the iliac crest and hip joint (Fig. 1A). An equally unique structure occurring only in hominins, the Anterior Inferior Iliac Spine (AIIS), is formed by an epiphysis that partially covers this growth plate at time of fusion in late adolescence. It is therefore a hominin synapomorphy (*Zirkle & Lovejoy, 2019*).

Corresponding authors
Dexter Zirkle, dzirkle1@kent.edu
C. Owen Lovejoy, Olovejoy@aol.com

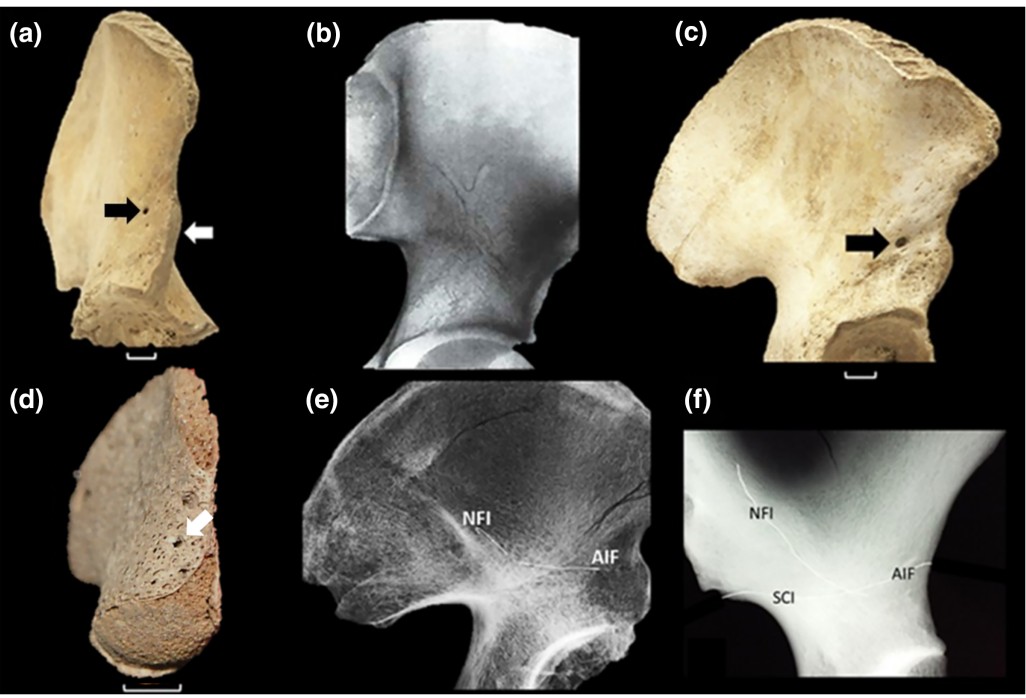

**Figure 1 The vascular canal system and associated traits.** (A) The ilium of a 12-year-old human with an open anterior physis (white arrow). At maturity, the physis becomes covered by a small, localized apophysis that forms the AIIS. A "bare area" (no physis) separates this future AIIS from the iliac crest. The portion of the physis between the AIIS and acetabular rim matures by direct ossification (no separate apophysis). This growth plate "complex" is absent in all other primates (*Zirkle & Lovejoy, 2019*). A black arrow identifies the specimen's AIF, which is very large. (B) Example of large vascular channels that are occasionally seen on X-ray, in this case in the shape of a large vertical "Y" (From *Köhler & Zimmer, 1968*, with permission). (C) 15-year-old with a large AIF (arrow). In most cases this foramen is significantly larger than the NFI which lies on the ilium's medial side. (D) 16-week fetal ilium with a large presumptive AIF already present (arrow), anterolateral view. (E) Demonstration of continuity of the AIF and nutrient foramen (NFI) in the ilium. A fine metal filament has been inserted into the AIF. A second has been introduced through the NFI. Such channels consistently meet at a central chiasma (for further discussion see text and (F). The two wires are touching but their contact is slightly obscured by the bone density at the point of intersection. Note that the wires have been clipped and bent at the point of their entrance in order to prevent loss into the cavity. (F) Male Gorilla os coxa (CMNH B1733). A fine metallic fiber can be seen to travel approximately 2/3 the breadth of the ilium before intersecting a second introduced through the NFI (unlike in (E) the wires were not clipped near their points of insertion). The approximate positions of three foramina are indicated. That marked "SCI" lies on the lateral iliac surface in *Gorilla* near the sciatic notch. It is a regularly occurring foramen. The NFI of gorillas lies on the lateral iliac surface. Human skeletal specimens are from the Libben Collection (*Lovejoy et al., 1977*).

The human ilium has also been demonstrated to contain a large central vascular canal system. Specifically, during X-ray and CT clinical examination, vascular "channels" of unusually large caliber are sometimes visible in patients. These can be mistaken for metastatic lesions or iliac fractures (Fig. 1B) (*Köhler & Zimmer, 1968*; *Richardson & Montana, 1985*; *Pikula, 1996*), and have been found to be a stable and consistent pathway through the ilium's cancellous bone (*Richardson & Montana, 1985*; *Pikula, 1996*; *Ebraheim et al., 1997*). Their distribution, caliber and contents were subjected to unusually detailed and extensive documentation by *Sirang (1973)*.

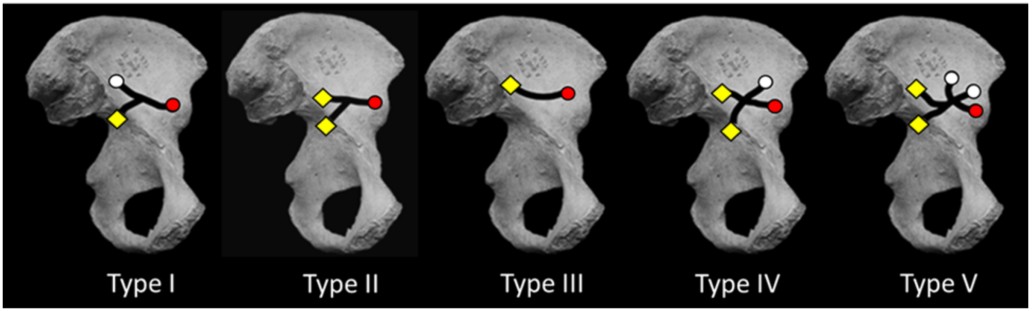

**Figure 2 Medial view of os coxa with iliac canal patterns demonstrated by Sirang (redrawn from _Sirang, 1973_).** Black lines indicate confirmed canal paths. Canals were consistently found to open at lateral (gluteal surface: white circle) and/or medial foramina (iliacus surface: yellow diamond) (cf. Fig. 1E). In this canal "system" the NFI was regularly the largest of the medial foramina, lying near the ilium's sacroiliac border. This is true except in Type I in which the NFI was found to open somewhat inferior to its more common position nearer the auricular surface. The AIF (red circle) is present in all cases reported by Sirang, and in all hominin ilia that we have examined. If present in other hominoids its lumen is rarely large enough to permit insertion of a fibrous probe. The Type III condition displays only a single canal connecting the NFI and AIF. For discussion see text.

We opined that if unique to hominins, the vascular canal system (hereafter the VCS) very likely developed and expanded in association with the emergence of its broad ilium and its unique anterior physis. We hypothesized that such a relationship is likely because the entrance to the canal system is one of the ilium's primary nutrient foramina and that the most prominent and consistent exit to the system lies in close proximity to the anterior physis when the latter is open. Moreover, that exit is universally present and frequently demonstrably large (Fig. 1C).

We explored whether such a system also characterizes the very differently shaped ilia of other primates, and whether such a system might be traceable in the human fossil record. However, because radiographic methods are ineffective in examining these passageways through iliac cancellous bone (_Richardson & Montana, 1985_; _Pikula, 1996_; _Ebraheim et al., 1997_), Sirang instead relied on extensive destructive dissection of a large sample of human ilia in order to map the VCS. As such sample sizes and destructive procedures are obviously not possible for highly valued non-human primate skeletal series, we sought another method by which to explore primate ilia.

In place of any destructive procedures, we explored the human VCS using fine-caliber probes introduced through the several regularly occurring and prominent foramina associated with it using Sirang's maps as a guide (Fig. 2). In doing so we discovered that one such opening, which we here refer to as the _anterior iliac foramen_ (or AIF) is consistently present in humans (Fig. 1C) and is demonstrably larger relative to body size than in all other primates that we examined, including _Pan_ and _Gorilla_. Indeed, our examination of other anthropoid ilia revealed no foramina that could serve as entrances or exits to a system similar to their role in the VCS.

We conclude that such an extensive canal system is unique to hominins and that other potentially homologous systems are either absent or lacking in sufficient volume to perform the same role of extensive vascular perfusion of the ilium during ontogeny.

**Table 1 Frequency of cartilage canal patterns from *Sirang (1973)*.**

| Type | Shape | Number Lateral Foramina | Number Medial Foramina | Freq (%) |
|---|---|---|---|---|
| I | "Y" | 2 | 1 | 31 |
| II | "Y" | 1 | 2 | 20 |
| III | Simple | 1 | 1 | 23 |
| IV | "X" | 2 | 2 | 15 |
| V | Complex | 3 | 2 | 11 |

Moreover, we also find it likely that the development of the VCS and its largest external vascular structure, the AIF, likely emerged early in the human fossil record in association with the hominin ilium's unique shape and breadth. Indeed, the ilia of *Australopithecus afarensis* (A.L. 288-1) and *Ardipithecus ramidus* (ARA-VP-6/500) both exhibit an AIF and one of demonstrably large caliber in the former. It would appear that the presence of an AIF is therefore a novel character useful in tracing the emergence of the bipedal ilium in fossil hominoids.

## MATERIALS & METHODS

Using blunt dissection, conventional X-ray, and CT of skeletons and cadavera of over 270 human adults (with and without contrast media), Sirang identified five distinct patterns in the VCS (Fig. 2; Table 1 and see below). He also injected media into the vascular contents of the system in fetuses and neonates. He "meticulously" (*Richardson & Montana, 1985*: 118) described the canals' contents and their regular anastomoses with vessels surrounding the perimeter of the ilium. His principal interest was the canals' role in venous drainage during radiographic examination using contrast media.

Sirang's Type III is a single canal that connects the most commonly identified nutrient foramen of the ilium (*Ebraheim et al., 1997*; *Cunningham & Black, 2013*; hereafter the NFI) to a solitary foramen lying close to the AIIS (Fig. 1C), and which we have identified as the AIF (see earlier). It is reasonable to presume that the contents of this most prominent canal are the primary vascular structures in the VCS, with other channels being occupied by less constant tributaries. Almost half of the patterns Sirang describes terminate exclusively at the AIF, and in human pelves that we have examined it regularly presents as a solitary and exceptionally large foramen (Fig. 1B). Its presence and unique amplitude can even be demonstrated early in fetal development (Fig. 1D).

Relying on Sirang's descriptions, published CT scans of living subjects, and detailed dissection of three human cadavera, we identified the most likely contents of the AIF to be either a subsidiary of the ascending branch of the Lateral Circumflex Femoral Artery and/or Vein and/or one from the Superior Gluteal Artery as likely routine arterial occupant(s) of the AIF (See S1).

To confirm the presence and continuity of the iliac canals in skeletal remains but without relying on any blunt or destructive procedures, we instead explored the VCS in human archaeological and museum specimens using slightly stiff but pliable, radiopaque

fibers of very fine caliber. We first explored the two paired foramina in Sirang's Type III (*i.e.*, the AIF and NFI).

We inserted a fiber into the AIF and found that it could easily be made to travel almost the entire breadth of the ilium without any significant resistance (*H. sapiens* = 38). We then inserted a second fiber into the NFI of the same specimen and found that it could also readily be made to travel a considerable distance, and one that was sufficient to allow it to physically intersect the first that was still resting in place from the AIF. This was confirmed by our ability to induce motion in either fiber by manipulating only the other (Fig. 1E). We radiographed a number of these specimens with the probes in place (Fig. 1E). Inasmuch as previous authors, and especially *Sirang (1973)*, had fully established the character and nature of the canal system in humans, we concluded that our method was satisfactory for testing for the presence of the canal system in other primates.

Our external examination of other primate ilia suggested that a similar system might be present in other hominoids based on the regular occurrence of a small AIF-positioned foramen in *Gorilla* and its frequent (but inconstant) presence in *Pan troglodytes* skeletons, but not in hylobatids, or in Old or New World monkeys as similar foramina in this region of the ilium are exceedingly rare and if occasionally present, exceptionally diminutive (see S2 & Figs. S2–S3). Moreover, we found no cases of bilateral occurrence in these specimens.

Our probe method proved unsuccessful in almost every hominoid that we tested (*N* = 32 for *Gorilla*; *N* = 28 for *Pan*; *N* = 6 for *Pongo*), usually because their foramina were not sufficiently ample to allow entrance and manipulation of our probes (see S3). However, in two gorilla specimens we were successful as shown in Fig. 1F (see S3). While a presumable AIF homologue might be typically present in some or even most African ape ilia, its caliber is apparently not sufficient to allow our manipulation test to succeed. Moreover, unlike the typical human state, the African ape NFI homologue typically lies on the ilium's external (dorsal) surface. As body size in *Pan* is often roughly similar to that of many human females and that of male gorilla is much larger than those of human males, we opined that body size was not a confounding issue.

Since our primary interest here was to confirm the presences of the canal system in fossil hominins, we hypothesized that an AIF of distinctly large caliber would be sufficient to establish its likely presence in human ancestral fossils. We therefore proceeded with a quantitative examination of the caliber of the AIF in humans and other hominoids.

We first measured the diameters of the AIF and NFI using dental calipers. While our metrics were reasonably reproducible (mean differences between first and second measurements equaled 3%), foramina were frequently angled and to some degree eccentric, making caliper placement sometimes problematic for guaranteed consistency, as most foramina are slightly ovoid with conically shaped reductions from the entrances.

To reduce this effect, we created a series of progressively size-incremented probes using "Tinkercad" software on a Formlabs Form2 3D stereolithography (SLA) printer (see S4 & Fig. S4). We inserted each probe into a target foramen in approximate alignment with its exit axis and allowed the probe to conform to any eccentricity of foramen shape. The "rule" for probe conformity with each foramen was to allow it to penetrate to a
depth of approximately 1.0 to 1.5 mm without resistance, *i.e.*, each probe was permitted to conform to slight variations in shape and angulation in a manner that did not prevent penetration to this approximate depth. In other words, the probe was allowed to penetrate the foramen's underlying canal just sufficiently to ensure that its largest dimension was being recorded. The largest probe in the series that a foramen would admit under these strictures was considered its diameter.

Single images from CT scans of two important hominin ilia, ARA-VP-6/500 and A.L. 288-1 were made available to us by Dr. Gen Suwa of the University Museum of the University of Tokyo. Dr. Suwa reports that these were taken using Skyscan 1,173 (Bruker) at 130 kV and 61 μA, with a 0.25 mm brass prefilter, at a voxel resolution of approximately 30 microns, and then visualized in Analyze software (Mayo Clinic) in 60-micron voxel resolution.

Finally, in order to correct for differences in body mass in our sample, we measured four dimensions that could also be used to construct a standardized size variable: femoral head diameter, humeral head diameter, distal humeral breadth, and proximal tibia breadth. We used the geometric mean of these four variables as a measure of body size in some of the calculations reported below.

For fossils, however, such composite sets of variables are only rarely available. We therefore also measured the vertical diameter of the acetabulum as a single indicator of body size. We measured it as the distance from the acetabular rim at the ischium along a line that also intercepted the anterior superior iliac spine (see *Krebs, Incavo & Shields, 2009*). In the single case reported below that has also been measured by other authors using different methods (A.L. 288-1) (*Hammond, Plavcan & Ward, 2013*), the two values for acetabular diameter are virtually identical.

## RESULTS

A regular presence and consistent location of an AIF is obvious in casts and photographs of original specimens of *Australopithecus afarensis* (Fig. 3A), *A. africanus*, and all examples of the genus *Homo* that we have examined thus far. The AIF diameters in specimens assigned to these taxa are listed in Table 2. The absolute and relative size of the NFI and AIF in humans, and other hominoids are shown in Table 3. An AIF is also present in the single known ilium of *Ardipithecus ramidus* (Figs. 3B–3E). Except for the *Ar. ramidus* specimen (ARA-VP-6/500), all have substantially greater normalized diameters than are typical of most chimpanzees, which are closest to these specimens both phylogenetically and in likely body mass (Figs. 4A, 4C & S7).

Our analyses of extant hominoid data are reported in Table 4 (see also tests of hypotheses and point and interval estimations of these ratios in S5 & Tables S1–S3). The mean human AIF diameter is significantly greater than are those of either *Pan* or *Gorilla* when corrected for body size, either normalized by vascular sufficiency, (AIF/NFI: Table 4) or divided by the Mosimann body size variable described earlier (*i.e.*, geometric mean) (Table 4). Moreover, eight of the nine values of the *Cohen's d* (mean difference divided by the pooled-within standard deviation) for these differences in ratios are highly significant, quite strong, and support the ordering of magnitudes: *Pan < Gorilla < Homo*.

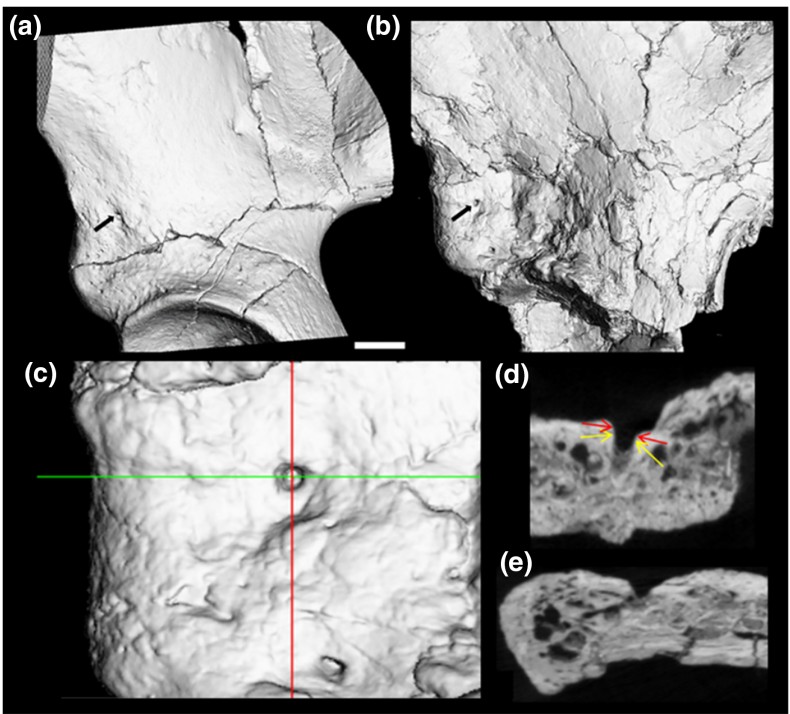

**Figure 3 CT render of AIF in two major hominin fossils.** (A) A.L. 288-1, (B) ARA VP-6/500. The AIF is denoted by small black arrow in each specimen. (B–E) ARA VP 6/500 (see Methods). (C) Major and minor axes of AIF in surface render. (D) Superoinferior section showing approximate points of measurement of diameter at about 0.5 mm depth (Yellow arrows: 1.1 mm), canal diameter is 1.0 to 1.1 mm to about a depth of 1.5 mm. Red arrows show approximate point of measurement of cast using dental calipers (for discussion see text). (E) Anteroposterior foramen section showing canal path near foramen.

**Table 2 AIF diameters in fossil hominins.**

| Fossil Taxa | Diameter (mm) | |
| --- | --- | --- |
| | Casts & Photos | CT scan |
| A.L. 288-1 | 1.8–1.9* | 1.85** |
| ARA-VP-6/500 | 1.3–1.4* | 1.1** |
| KSD-VP-1/1 | 1.8* | |
| Sts 14 | 2.5* | |
| SK 3155 | 2.0* | |
| MLD 7 | 2.3* | |
| MLD 25 | 1.6* | |

Notes:
* All metrics are estimates either from casts and/or photographs of original specimens. Any conclusions with respect to fossil specimens must therefore await data acquired from further examination of these specimens.
** Diameters were obtained at different depths within foramen

The single exception is the AIF/NFI measure, which detects no difference between *Pan* and *Gorilla* (Table 4). Together, these data indicate that AIF diameter is primitive in African apes but robustly derived in humans and earlier hominins (see below).

**Table 3 Means of AIF, NFI, and individual paired differences in adult hominoids (mm), with standard deviations, t-values (and df), and *Cohen's d* effect sizes.**

| Taxon | AIF | NFI | AIF-NFI | s* | t(df) | *Cohen's d* |
|---|---|---|---|---|---|---|
| *Homo sapiens* | 2.29 | 1.47 | 0.82 | 0.80 | 8.26 (64) | 1.02** |
| *Pan troglodytes* | 1.03 | 1.02 | 0.01 | 0.64 | 0.09 (45) | 0.01 ns |
| *Gorilla gorilla* | 1.88 | 1.71 | 0.17 | 0.73 | 1.58 (47) | 0.23 ns |

**Notes:**
* Standard deviation of the paired differences.
** highly significant, 0.01.
ns, nonsignificant.

We estimated (using dental calipers) the entrance to the AIF of ARA-VP-6/500 to be 1.3 to 1.4 mm. We did so while attempting to duplicate use of our specially designed probe (see earlier). However, CT data suggest that the actual diameter of the AIF in this specimen is best estimated as 1.1 mm at a depth of between 0.5 and 1.5 mm from the iliac surface, and our probe is normally inserted to a depth of about 1 mm, but this is, of course, always an estimate (see Methods). However, as can be seen from the CT image provided by Prof. Suwa (Figs. 3B–3E), the largest dimension of the obviously flared entrance to the AIF of this specimen is, in fact, quite close to our original estimate, although without actual application of our method directly to the original specimen, the comparability of these data must remain provisional.

The picture is much clearer in the case of A.L. 288-1 ("Lucy") for which Prof. Suwa also supplied a CT image (Fig. 3A). This specimen is largely intact and has not suffered as much damage as has the *Ar. ramidus* ilium, although it appears to still have contained some matrix at the time of scanning (White and Suwa, personal communications). Nevertheless, an estimate of its AIF diameter from our CT image (1.85 mm) is squarely within the range of the values we estimated from our Cleveland Museum of Natural History casts using dental calipers and photographs of the original (1.80–1.90 mm). Moreover, the acetabulum of A.L. 288-1 is well preserved, and when used to normalize its AIF diameter, the resulting index falls well above the 75th percentile of all other hominoids and even at the high end of those of humans (Fig. 4B). It is notable that several other hominin fossils also appear to exhibit large AIFs, including Sts 14, SK 3155 (see S6 & Fig. S5), OH 28, and KNM-ER 3228. We posit that a reexamination of hominin fossil pelvic remains is warranted considering the clear importance of this trait.

## DISCUSSION

The cross-sectional area of a bone's nutrient (or other penetrating) foramina is related to "blood flow requirements of the internal bone cells that are essential for dynamic bone remodeling" (*Seymour et al., 2012*: 451). Flow resistance in arterial vessels is linearly increased in relation to their length (a derivation from Poiseuille's equation), a parameter that is obviously dictated by anatomical "requirements", *i.e.*, the distance between the structure and the organism's central vascular tree. However, positive flow in any vessel is proportional to the fourth power of its radius (also a derivation from Poiseuille's equation (*Seymour et al., 2012*)). Therefore, a functional increase in flow necessary for the

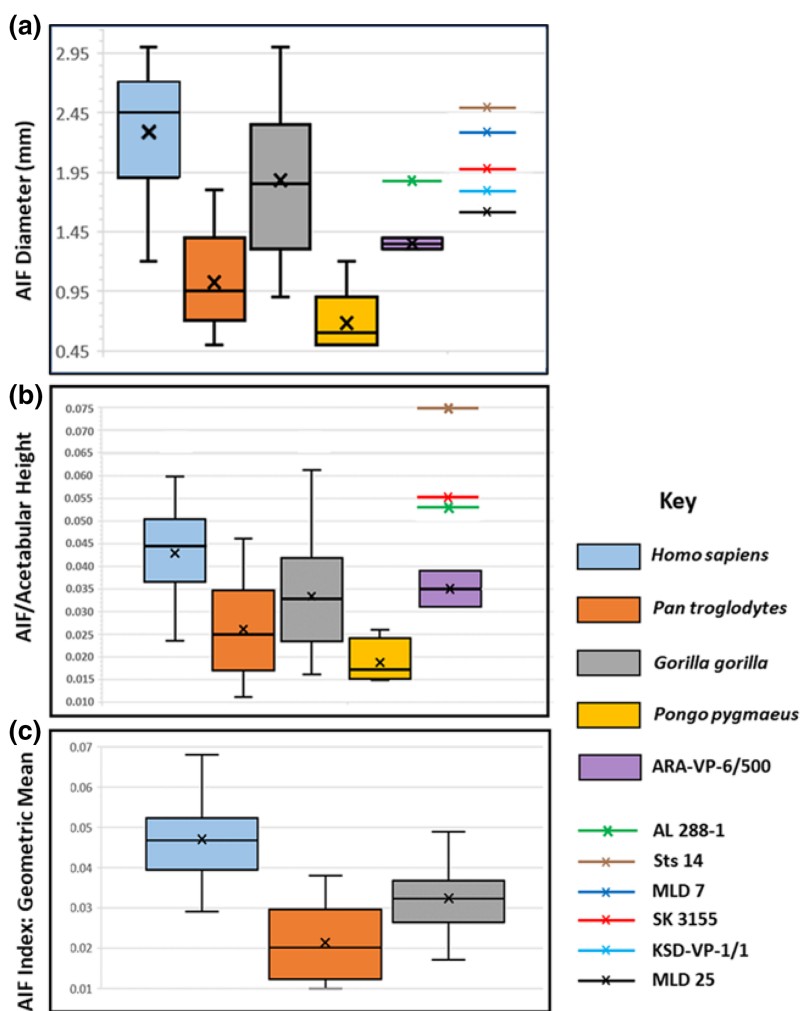

**Figure 4 Maximum diameter of AIF (see Methods) in extant hominoids and fossil hominins.** (A) Raw data from casts, photographs of originals and CT scans where available (see text). (B) AIF diameter/acetabular height. Maximum and minimum estimates of acetabular height for A.L. 288-1 (*Johanson et al., 1982*) and ARA-VP-6/500 (*Lovejoy et al., 2009*) have been used for size normalization factors. Acetabular diameters for Sts 14 and SK 3155 from (*Hammond, Plavcan & Ward, 2013*). Highest value for ARA-VP-6/500 is probable maximum foramen diameter divided by minimum estimated acetabular height, whereas lowest value is minimum probable foramen diameter divided by maximum estimated acetabular height. (C) AIF/geometric mean of four skeletal sites (see text). Humans differ significantly from *Pan* and *Gorilla*. Extant taxa boxes are 25th and 75th percentiles, whiskers are observed range, and transverse lines are median values. Box height is the interquartile range. All data presented here for fossil specimens are suggestive, as they have not yet been confirmed using the techniques described in Methods for non-fossil taxa.

**Table 4 *Cohen's d* effect size for each ratio.**

|  | AIF/NFI | AIF/Acet. | AIF/Geom. Mean |
|---|---|---|---|
| *H. sapiens/P. troglodytes* | 0.791** | 1.783** | 2.555** |
| *H. sapiens/G. gorilla* | 0.854** | 0.943** | 1.582** |
| *P. troglodytes/G. gorilla* | 0.038 ns | 0.698** | 1.306** |

**Notes:**
** highly significant at 0.01.
ns, nonsignificant.

development of a novel physis can be expected to substantially increase the radius of the lumina of the vessels responsible for both the inflow and outflow of blood sufficient to provide enhanced nutrient and waste flow during growth. Therefore, dimensions of nutrient foramina can serve as indicators of relative metabolic intensity (*Hu, Nelson & Seymour, 2020*; *Seymour et al., 2012*; *Houssaye & Prévoteau, 2020*). Inasmuch as the AIF is unusually large in hominins and is intimately related to other foramina *via* the VCS, we hypothesize that its extraordinary size and regular occurrence are unique attributes directly associated with the origin and evolution of the anterior physis synapomorphy (*Zirkle & Lovejoy, 2019*). The only alternative would be to argue that the enlarged AIF uniquely supplies the hominin acetabulum, but the acetabulum of early hominins is not enlarged as it is in *Homo* (*Lovejoy, 2005*), and the presence of such an enlarged AIF in A.L. 288-1 does not support such a hypothesis since its acetabulum is also distinctly smaller than in more recent human homologues (*Johanson et al., 1982*; *Lovejoy, 2005*).

It is well established that the expansive breadth and reorientation of the hominin ilium repositions the anterior gluteals for pelvic support during stance phase (*Aiello & Dean, 1990*; *Berge, 1994*; *Gruss & Schmitt, 2015*; *Le Gros Clark, 1964*; *Lovejoy, 2005*; *Lovejoy, 1988*; *Ruff, 1998*; *Ruff, 1991*; *Stern, 1988*). It has also previously been demonstrated that the human anterior physis is a unique modification that is integral to changes in iliac structure associated with bipedality, as the physis greatly expands iliac isthmus size and thus alters the orientation of the anterior gluteals (*Zirkle & Lovejoy, 2019*). Based on the findings outlined above we hypothesize that a significant intensification of the vascular supply to the physis, including both the ingress and egress of blood, resulted in either the initial origin of the hominin ilium's expanded canal system or alternatively, its unique expansion if such a system was already present in primitive form. Given either of these possibilities, the facts that (1) a foramen is occasionally located in the same position as the human AIF but is absent in some African ape individuals (especially chimpanzees); and (2) if present, the foramen is no larger than its nutrient foramen (see Results); and (3) that we were able to demonstrate a possible homologous system in only two of 32 total gorilla specimens, all suggest that the marked breadth of the hominin iliac isthmus required a substantial increase in the diameter of the ilium's primary vascular tree, which in turn resulted in a substantially enlarged AIF.

This being the case, it would seem that the presence and significant enlargement of an AIF in a fossil ilium can reasonably serve as an indicator that its AIIS emerged *via* the unique process found only in hominins. Such a conclusion is consistent with the specialized nature of AIIS morphology that is also clearly present in hominins (*Lovejoy et al., 2009*), especially its greatly expanded isthmus breadth. Current evidence suggests that both a significantly expanded acetabular growth plate and an accompanying increased blood supply during development, were present in hominins by 3.2 My (in A.L. 288-1), and were likely present by 4.4 My (*i.e.*, ARA-VP-6/500), although the latter must await further examination of presently available specimens as well any yet discovered. It is unfortunate that, to our knowledge, no currently known Mio-Pliocene fossil hominoid ilia are sufficiently preserved to allow accurate observation of the region of the anterior foramen. This includes the singular known iliac specimen from *Oreopithecus* as it neither

exhibits any evidence of an anterior physis like that seen in hominins nor is sufficiently preserved to permit informative observation of its surface. We obviously applaud current efforts to amplify our lack of fossils from this critical period of hominin evolution.

The presence of this unusual synapomorphy in all known hominin ilia subsequent in age to 4.4 My robustly suggests that habitually upright, non-saltatory, terrestrial bipedality was a unique event, having occurred only once in the primate fossil record. Current and future efforts to identify potential cladistic ancestral relationships of known hominins using only osteometric and spatial data must therefore be appropriately modified to emphasize the unique ontogeny, vascularity, and morphology of the anterolateral ilium.

## ACKNOWLEDGEMENTS

We thank Tim D. White and Gen Suwa for examining original specimens and especially for obtaining CT scans, photographs, and metrics, which have proved integral to the successful completion of this work. We thank Yohannes Haile–Selassie formerly of the Cleveland Museum of Natural History for access to skeletal materials. We thank Melanie McCollum for helpful discussions. Thanks to Brian Grafton for generous use of the Kent State University human gross anatomy lab. Lastly, thanks to Kent State University's Spark Innovation Studio for use of the Formlabs Form2 3D photopolymer resin printer.

### Funding
The authors received no funding for this work.

### Competing Interests
The authors declare that they have no competing interests.

### Author Contributions
- Dexter Zirkle conceived and designed the experiments, performed the experiments, analyzed the data, prepared figures and/or tables, authored or reviewed drafts of the paper, and approved the final draft.
- Richard S. Meindl conceived and designed the experiments, analyzed the data, prepared figures and/or tables, authored or reviewed drafts of the paper, and approved the final draft.
- C. Owen Lovejoy conceived and designed the experiments, analyzed the data, prepared figures and/or tables, authored or reviewed drafts of the paper, and approved the final draft.

### Data Availability
The data are available at Open Science Framework:

-https://osf.io/kt5cq.

-https://osf.io/j9c8s.

Dr. Gen Suwa's contact information: The University Museum, the University of Tokyo, Hongo, Bunkyo–ku, Tokyo 113-0033, Japan.

## Supplemental Information

Supplemental information for this article can be found online at http://dx.doi.org/10.7717/peerj.12240#supplemental-information.

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
