# Peer review of "Upright walking has driven unique vascular specialization of the hominin ilium"

_PeerJ, doi:10.7717/peerj.12240_

## Round 0.1 · original submission · Minor Revisions

Sorry for the delay, but I was waiting on a third review that never materialized, so I have stepped in to add comments to the manuscript (see attached). Regardless, you can see that the two reviews I did receive are highly favorable of your work, essentially arguing for acceptance. I have marked this "Minor Revisions" because Reviewer 2 makes valid though minor suggestions. In addition, please realize that PeerJ does not make use of a copy editing service, so please do your best to check your revised manuscript as closely as possible for formatting and grammar. I have indicated a few examples in the attached pdf. I look forward to receiving your revised manuscript.

Reviewer 1 ·

Basic reporting

Excellent. Superbly well-written and well-researched (and see Additional Comments)

Experimental design

Excellent in all ways (and see Additional Comments)

Validity of the findings

Clearly stated in the text. All aspects are excellent.

Additional comments

Few authorities in the human/hominin osteology (including me) have likely made much note of the presence of an enlarged, anteriorly-positioned foramen on the lateral face of the human ilium, let alone in early fossil hominins. In fact, as I began reading this manuscript, I looked at human and nonhuman primate ilia in our collection and had the same “ah-hah” moment that Zirkle and collaborators must have had when making the discovery. In this well-written and hypothesis-driven manuscript, Zirkle and co-authors lead the reader through the study of the vascular specialization and interpretation of the morphology of the foramen. They convincingly argue that the specialization is a central element of the enlarged central vascular canal system that is unique to hominins, including in the earliest hominins (Au. afarensis and Au. ramidus). They make the clear case that the largest external vascular structure (AIF) appears first in the hominin fossil record as a part of the unique morphology associated with bipedality.
I am impressed with the detail of the science behind the study, including a compelling comparison based on the record of dissection, radiography, and computed tomography of human adults produced by earlier researchers. Zirkle and collaborators thorough study provides a comparison of the foramen’s morphology in archaeological materials, CT scans, casts of early hominins, photographs, dissections, and descriptions from the literature. In addition to measurements of the feature, images from CT scans of two Ardipithecus ilia and correction for body size differences firmly establishes the uniqueness of the foramen exclusive to extant and extinct hominins. The investigators suggest that the enlarged AIF and morphology is associated with increase in blood supply during development.
In summary, this is an extraordinarily well written article, test of a compelling hypothesis, and documentation of a clear association between morphology and behavioral adaptations surrounding bipedality. I strongly recommend publication. In my view, aside from some minor editorial revisions (e.g., upper case for the journal title for the Hammond et al. reference, italics for the journals Science, Skeletal Radiology, and Clinical Orthopaedics, italics for Ardipithecus ramidus, and other minor revisions, the manuscript is ready for publication.

Reviewer 2 ·

Basic reporting

The paper is clear, concise, and easy to follow. There are a few places where additional citations and word changes are important.

It would be delightful if some specimens prior to Ardipithecus were discussed--Oreopithecus?? What might be happening during the Miocene that helps us discern the likelihood that an arboreal biped and/or flexible scrambler already had an additional vascular network in the ilium? I understand leaving the discussion of other, later (South African) material to be tested by others, but I think earlier material does need to be included here, at least in discussion.

Line 243, please cite something other than the Zirkle & Lovejoy paper here. The idea of using the glutes during stance phase is hugely important in biomechanics and having an actual biomechanics study (EMG?) cited here will help broaden the interest and background for the vascularization argument put forth by these specific authors.

Most importantly, I am not sure why the authors refer to hominin bipedalism as cursorial (in the final paragraph). While I understand that cursorial can simply mean continuous, or endurance, it will typically be used to imply running; the inference that all forms of our lineage allowed for running does not stem from this study or any data of which I am aware. To this end, another word here is quite important.

Experimental design

Appropriate design for the study. Clear, relevant.

Validity of the findings

No comment

---

## Round 0.2 · accepted · Accept

Thank you very much for considering the reviewers' critiques and suggestions. The paper is ready to be accepted. Once it goes to production things move pretty fast. When you get the proofs back that is the last time to correct any typographical errors, please review it closely.

Congratulations on an insightful paper.